

# Acclimation of liverwort *Marchantia polymorpha* to physiological drought reveals important roles of antioxidant enzymes, proline and abscisic acid in land plant adaptation to osmotic stress

Totan Kumar Ghosh[1], Naznin Haque Tompa[1], Md. Mezanur Rahman[2], Mohammed Mohi-Ud-Din[1], S. M. Zubair Al-Meraj[1], Md. Sanaullah Biswas[3] and Mohammad Golam Mostofa[2,4]

[1] Department of Crop Botany, Bangabandhu Sheikh Mujibur Rahman Agricultural University, Gazipur, Bangladesh
[2] Institute of Genomics for Crop Abiotic Stress Tolerance, Department of Plant and Soil Science, Texas Tech University, Lubbock, Texas, United States
[3] Department of Horticulture, Bangabandhu Sheikh Mujibur Rahman Agricultural University, Gazipur, Bangladesh
[4] Department of Biochemistry and Molecular Biology, Bangabandhu Sheikh Mujibur Rahman Agricultural University, Gazipur, Bangladesh

Corresponding author
Mohammad Golam Mostofa,
mmostofa@ttu.edu

## ABSTRACT

Liverwort *Marchantia polymorpha* is considered as the key species for addressing a myriad of questions in plant biology. Exploration of drought tolerance mechanism(s) in this group of land plants offers a platform to identify the early adaptive mechanisms involved in drought tolerance. The current study aimed at elucidating the drought acclimation mechanisms in liverwort's model *M. polymorpha*. The gemmae, asexual reproductive units of *M. polymorpha*, were exposed to sucrose (0.2 M), mannitol (0.5 M) and polyethylene glycol (PEG, 10%) for inducing physiological drought to investigate their effects at morphological, physiological and biochemical levels. Our results showed that drought exposure led to extreme growth inhibition, disruption of membrane stability and reduction in photosynthetic pigment contents in *M. polymorpha*. The increased accumulation of hydrogen peroxide and malondialdehyde, and the rate of electrolyte leakage in the gemmalings of *M. polymorpha* indicated an evidence of drought-caused oxidative stress. The gemmalings showed significant induction of the activities of key antioxidant enzymes, including superoxide dismutase, catalase, ascorbate peroxidase, dehydroascorbate reductase and glutathione *S*-transferase, and total antioxidant activity in response to increased oxidative stress under drought. Importantly, to counteract the drought effects, the gemmalings also accumulated a significant amount of proline, which coincided with the evolutionary presence of proline biosynthesis gene $\Delta^1$-*pyrroline-5-carboxylate synthase 1* (*P5CS1*) in land plants. Furthermore, the application of exogenous abscisic acid (ABA) reduced drought-induced tissue damage and improved the activities of antioxidant enzymes and accumulation of proline, implying an archetypal role of this phytohormone in *M. polymorpha* for drought tolerance. We conclude that physiological drought tolerance mechanisms governed by the cellular antioxidants, proline and ABA were

adopted in liverwort *M. polymorpha*, and that these findings have important implications in aiding our understanding of osmotic stress acclimation processes in land plants.

# INTRODUCTION

Due to constant threats of unprecedented global warming and environmental deterioration, plants are more frequently exposed to a variety of abiotic stresses during their life cycle (*Godoy et al., 2021*). Being sessile in nature, land plants have to confront a lot of challenges for their survival and acclimatization to environmental stresses. Drought has been considered as one of the major abiotic constraints, reducing crop productivity worldwide (*Lesk, Rowhani & Ramankutty, 2016*; *Iqbal, Singh & Ansari, 2020*). During adaptation to drought, land plants trigger several morphological changes, such as reduction of leaf area and modification of stem structures to efficiently adjust to less availability of resources. Along with morphological adaptation, they also modify various physiological, biochemical and molecular processes to survive under drought conditions (*Lim et al., 2015*; *Sharma et al., 2019*). For example, land plants reduce water loss by stimulating stomatal closure in response to drought stress (*Lim et al., 2015*). Accelerated production of reactive oxygen species (ROS), such as super oxide ($O_2^{\bullet-}$), hydrogen peroxide ($H_2O_2$) and hydroxyl radical ($^{\bullet}OH$) are remarkable feature of drought-induced adverse effects in plants (*Ahmad & Umar, 2011*; *Dumont & Rivoal, 2019*). Increased accumulation of ROS causes damage to DNA, RNA, protein and lipid membrane, resulting in cellular injury and even death of cells (*Gapper & Dolan, 2006*; *Raja et al., 2017*). Plants are also equipped with a vibrant antioxidant system, comprising of both enzymes and non-enzymatic compounds, to protect ROS-induced oxidative damage under drought conditions (*Laxa et al., 2019*).

Alongside, during acclimation to drought stress, plants rapidly accumulate different types of osmoprotectants to reduce the potential hazards of drought to cell membranes and organelles. The osmoprotectants, including sugars, amino acids and polyols play a pivotal role in stabilizing stress-responsive proteins, scavenging ROS and adjusting osmotic potential (*Kumar et al., 2018*). Among the osmoprotectants, proline has been widely accepted as a stress-marker and reported to have beneficial roles in safeguarding plants from the negative impacts of drought stress (*Shinde et al., 2016*; *Gurrieri et al., 2020*; *Abdelaal et al., 2020*). In fact, drought leads to an enhanced expression of proline biosynthesis gene $\Delta^1$-*pyrroline-5-carboxylate synthetase1* (*P5CS1*), leading to the accumulation of proline in cellular compartments (*Furlan et al., 2020*). Besides, the phytohormone abscisic acid (ABA) plays a paramount role in mounting drought tolerance mechanism in plants. The drought-induced ABA can enhance the functions of cellular antioxidants and osmolytes for scavenging ROS and osmotic adjustment, respectively,

thereby protecting cellular machineries from osmotic and oxidative stresses (*Shinde, Islam & Ng, 2012*; *Skowron & Trojak, 2021*). In contrast to angiosperms, the evolutionarily important land plants like bryophytes have been less explored to identify their drought acclimation mechanisms driven by the antioxidant defense system, osmolyte accumulations and ABA.

Bryophytes, which include mosses, liverworts, and hornworts have recently been resolved as the monophyletic group that occupies a crucial position in land plant phylogeny (*Puttick et al., 2018*; *Delaux et al., 2019*; *Cheng et al., 2019*; *Zhang et al., 2020*; *Harris et al., 2020*; *Su et al., 2021*). During colonization of lands by streptophyte algae, the common ancestors of the land plants and their closest bryophytes were challenged by severe environmental factors, including water shortage. The attributes of vegetative desiccation tolerance are rather common in bryophytes but have been lost in vascular plants during evolution (*Oliver, Velten & Mishler, 2005*; *Proctor et al., 2007*). Therefore, physiological, biochemical and molecular mechanisms associated with the drought tolerance in bryophytes should be an essential aspect of plant stress biology to comprehend the evolutionary processes of drought tolerance in this group of land plants. As compared to mosses, the early adaptive mechanisms of liverworts in response to drought are poorly understood. Among the bryophytes, liverwort *M. polymorpha* has been widely used as the model system to elucidate the evolutionary aspects of the stress tolerance mechanisms in land plants (*Ghosh et al., 2016*; *Bowman et al., 2017*; *Sugano et al., 2018*; *Jahan et al., 2019*; *Godinez-Vidal et al., 2020*; *Wu et al., 2021*). *M. polymorpha* possesses a dorsiventral thallus body, lacking vascular tissues and true roots but collectively possess unique features for plant biological study (*Shimamura, 2016*). The asexual reproductive units (gemmae) of *M. polymorpha* exhibit excellent drought tolerance potential (*Akter et al., 2014*); however, the putative drought acclimation mechanisms in gemmae are yet to be fully explored. Although drought-induced oxidative stress, antioxidant responses and osmolyte accumulations were investigated in mosses (*Hu et al., 2016*; *Cruz de Carvalho et al., 2017*; *Pizarro et al., 2019*), the information of ROS generation and its detoxification system, and osmolyte-regulated drought tolerance are largely unknown in liverworts. Moreover, the role of ABA in the accumulation of compatible solutes and ROS-detoxification systems under osmotic stress are still to be clarified in liverworts. Considering the above facts, the present efforts were made to investigate the physiological drought responses in liverwort *M. polymorpha* by exposing them to osmotic stress inducers, including sucrose, mannitol and polyethylene glycol (PEG). We explored several drought tolerance indicators like antioxidant defense, osmoprotectant accumulations and ABA responsiveness in the gemmae of *M. polymorpha* to understand the osmotic stress acclimation processes in liverworts.

# MATERIALS & METHODS

## Culture of plant materials, growth conditions and treatments

The gemmae of *M. polymorpha* (Takaragaike 1; TAK 1) were cultured in Gamborg's ½ B5 solid medium by following the standard growth conditions as described previously (*Akter et al., 2014*). After 15 days of solid culture, a portion of thallus body was harvested,

and the fresh weight (initial weight) was immediately recorded using a digital balance. The weighted thallus bodies were then cultured in ½ B5 solid medium (control) or in ½ B5 solid medium supplemented with 0.2 M sucrose, 0.5 M mannitol and 10% polyethylene glycol (PEG 6,000) for inducing osmotic stress (*Akter et al., 2014*; *Takezawa et al., 2015*; *Basal, Szabo & Veres, 2020*). To analyze the growth performance, the appearance of thallus growth was photographed, and the thallus weight (final weight) was recorded after 15 days of exposure to drought stress. The thallus weight-gain was calculated by deducting initial weight from final weight. For physiological and biochemical assays, liquid culture of the gemmae was carried out following the aforementioned growth conditions with continuous agitation at 130 rpm for 3 days (*Ghosh, Kaneko & Takezawa, 2016*). The gemmaling were then cultured in either ½ B5 liquid medium (control) or ½ B5 liquid medium containing 0.2 M sucrose, 0.5 M mannitol and 10% PEG for another 2 days. For investigating ABA responsiveness, the gemmalings were exposed to 10 μM ABA containing ½ B5 liquid medium for a period of 2 days (*Ghosh, Kaneko & Takezawa, 2016*; *Jahan et al., 2019*). The treated and untreated gemmalings were collected and stored at −20 °C for analyses of various physiological and biochemical parameters.

## Determination of electrolyte leakage and chlorophyll content

The electrolyte leakage in the control and drought-treated gemmalings was quantified following the detailed procedure reported by *Nagao et al. (2005)* using a conductivity meter (LAQUAtwin EC-11, Japan). The contents of chlorophyll (Chl) $a$, Chl $b$ and Chl $(a + b)$ in the control and osmotic stress-treated gemmalings were determined and calculated according to the report of *Porra (2002)*.

## Determination of proline content

Proline extraction and determination were carried out following the procedures reported by *Bates, Waldren & Teare (1973)*. Briefly, gemmalings (0.1 g) were homogenized in 2.5 mL of 6% aqueous sulfosalicylic acid followed by centrifugation at 4,000 rpm for 20 min. A volume of one mL of each supernatant was mixed with one mL of acid ninhydrin and one mL of glacial acetic acid. After heating in a boiling water bath for 60 min, the tubes were immediately transferred to an ice bath to terminate the reaction. Afterward, two mL of toluene was added to the reaction mixture and kept at room temperature for 10 min. Finally, the absorbance was measured spectrophotometrically at 520 nm using toluene as a blank. The level of proline in the samples was calculated from a standard curve developed with different concentrations of proline.

## Phylogenetic analysis of proline biosynthesis gene

To see the evolutionary relationship of proline biosynthesis gene, the phylogenetic analysis was done using proline biosynthesis enzyme (P5CS1) in the representative models of land plants such as bryophytes (Mosses *Ceratodon purpureus*, *P. patens*, *Sphagnum fallax*, and *Sphagnum magellanicum*, liverwort *M. polymorpha*), pteridophyte (*Ceratopteris richardii*), lycophyte (*Selaginella moellendorffii*), and angiosperms (*Arabidopsis thaliana*, *Oryza sativa* and *Populus trichocarpa*). The protein sequences were collected by BLAST

search using the databases of Phytozome v13. The amino acid sequences were aligned by ClustalW program. The phylogenetic tree was built by MEGA5 program (*Tamura et al., 2011*) using the Maximum Likelyhood method based on the Jones-Tailor-Thronton (JTT) matrix-based model (*Jones, Taylor & Thornton, 1992*). The tree followed bootstrap method with 1,000 replication (*Felsenstein, 1985*). All positions containing gaps and missing data were eliminated. The number on the branches represented bootstrap values. The number of substitutions per site was indicated by a scale bar.

## Determination of hydrogen peroxide ($H_2O_2$) and malondialdehyde (MDA) contents

For the determination of hydrogen peroxide ($H_2O_2$) content, the extraction of $H_2O_2$ from gemmalings (0.1 g) was carried out using 0.1% trichloroacetic acid (TCA) solution. After centrifugation at 12,000 rpm for 15 min at 4 °C, the supernatant was mixed with potassium iodide (1.0 M) and KP buffer (10 mM, pH 7.0) solution followed by incubation at room temperature for 30 min. Finally, the absorbance of the mixture was read at 390 nm, and the content of $H_2O_2$ was measured as $\mu$mol $g^{-1}$ FW (*Siddiqui et al., 2021*). The amount of MDA was determined following the procedure reported by *Siddiqui et al. (2021)* with slight modification. Briefly, the gemmalings (0.1 g) were crushed in 0.1% TCA followed by centrifugation at 11,500 rpm for 15 min at 4 °C. The resultant supernatants were mixed with thiobarbituric acid (TBA) solution (0.5%, prepared in 20% TCA). The resultant solution was heated in a boiling water bath for 30 min at 95 °C, then cooled quickly in an ice bath to terminate the reaction. Finally, the absorbance of the mixture was read at 532 nm and the content of MDA was measured as nmol $g^{-1}$ FW.

## Analysis of antioxidant enzymes and total antioxidant activity

The extraction of total soluble proteins and the preparation of supernatants for enzyme assay were done according to the comprehensive procedures reported by *Mostofa et al. (2020)*. Briefly, gemmalings (0.1 g) were crushed in ice-cooled mortar and pestles using an extraction buffer comprised of 50 mM ice-cold KP buffer (pH 7.0), potassium chloride (100 mM), ascorbate (AsA, 1 mM), $\beta$-mercaptoethanol (5 mM) and glycerol (10%, v/v). The homogenates were centrifuged at 11,500 × g for 12 min at 4 °C, and the supernatant was collected for quantifying soluble protein content following the standard protocol of *Bradford (1976)*, as well as for estimating different enzyme activities. The superoxide dismutase (SOD) activity was expressed as unit $mg^{-1}$ protein (inhibition of nitroblue tetrazolium reduction by 50% per minute) using xanthin-xanthin oxidase system by following the protocol of *Beyer & Fridovich (1987)*. The standard protocol of *Aebi (1984)* was followed for assessing the activity of catalase (CAT) and it was expressed as $\mu$mol $min^{-1}$ $mg^{-1}$ protein. The activity of ascorbate peroxidase (APX) was determined according to the protocol used by *Nakano & Asada (1981)* and it was expressed as $\mu$mol $min^{-1}$ $mg^{-1}$ protein. The activity of dehydroascorbate reductase (DHAR) was assessed according to the protocol described by *Nakano & Asada (1981)* and it was expressed as nmol $min^{-1}$ $mg^{-1}$ protein. The method for glutathione *S*-transferase (GST) activity determination was adopted from *Hossain, Hasanuzzaman & Fujita (2010)* and expressed

as nmol min$^{-1}$ mg$^{-1}$ protein. The total antioxidant capacity (TAC) of the methanolic extracts of gemmalings were determined by measuring the 2, 2-diphenyl-1-picrylhydrazyl (DPPH) scavenging activity (%) using the method reported by *Xu et al. (2010)*.

### ABA-induced drought tolerance assay

To observe the efficacy of ABA in improving the tissue survivability under physiological drought, the gemmalings pretreated with or without ABA (10 μM) for 2 days were exposed to 0.2 M sucrose, 0.5 M mannitol and 10% PEG for another 2 days. Afterword, the gemmalings were stained with Evan's blue dye solution (1% w/v) following the method reported by *Takezawa et al. (2015)*. The tissue survivability in the gemmalings was observed with the help of a light microscopy system (ZEISS Primostar, Germany).

### Statistical analysis

One way analysis of variance (ANOVA) was carried out for the obtained data using the Statistix 10 software package. To determine the significant differences among the treatments, the post-hoc test, namely least significant difference (LSD) at $P < 0.05$, as well as the student $t$-test, were used. The numerical values were presented in the figures as means ± standard errors (SEs) of three independent replications.

# RESULTS

## Drought-induced adverse effects on growth and photosynthetic pigments in *M. polymorpha*

During acclimation to drought stress, land plants follow numerous changes in their morphology like reduction of vegetative growth to limit water lose from their body. Osmotic stresses induce morphological growth-arrest along with the reduction of Chl contents in angiosperms, including rice and *Arabidopsis* (*Ozfidan et al., 2013*; *Kumari et al., 2014*; *García-Morales et al., 2018*). Exposure of *M. polymorpha* thallus to the osmotic stressors for 15 days resulted in severe growth inhibition (Fig. 1A). In comparison with stress-free control, the thallus weight-gain was decreased by 91.98%, 94.84% and 52.72% upon exposure to 0.2 M sucrose, 0.5 M mannitol and 10% PEG, respectively. The reduction in thallus weight-gain was coincided with the recorded phenotypes of *M. polymorpha* thallus under osmotic stresses (Figs. 1A & 1B). Maintenance of photosynthetic efficiency is a big challenge for land plant during acclimation to any abiotic stresses, including drought (*Sharma et al., 2020*). To better understand the drought-caused negative effects on photosynthetic pigments, we measured the levels of Chl *a*, Chl *b* and Chl (*a* + *b*) in gemmalings of *M. polymorpha*. Osmotic stresses induced by 0.2 M sucrose, 0.5 M mannitol and 10% PEG decreased the contents of Chl *a* (by 72.35%, 81.48% and 50.64%), Chl *b* (by 64.86%, 78.33% and 41.70%) and Chl (*a* + *b*) (by 69.05%, 80.10% and 46.72%), respectively in gemmalings of *M. polymorpha*, when compared with those levels of stress-free gemmalings (Figs. 1C–1E). These results implied that liverwort *M. polymorpha* exhibited growth defects coupled with photosynthetic pigments degradation in response to physiological drought, as commonly observed in angiosperms under drought conditions (*Zhao et al., 2020*; *Hossain et al., 2020*).

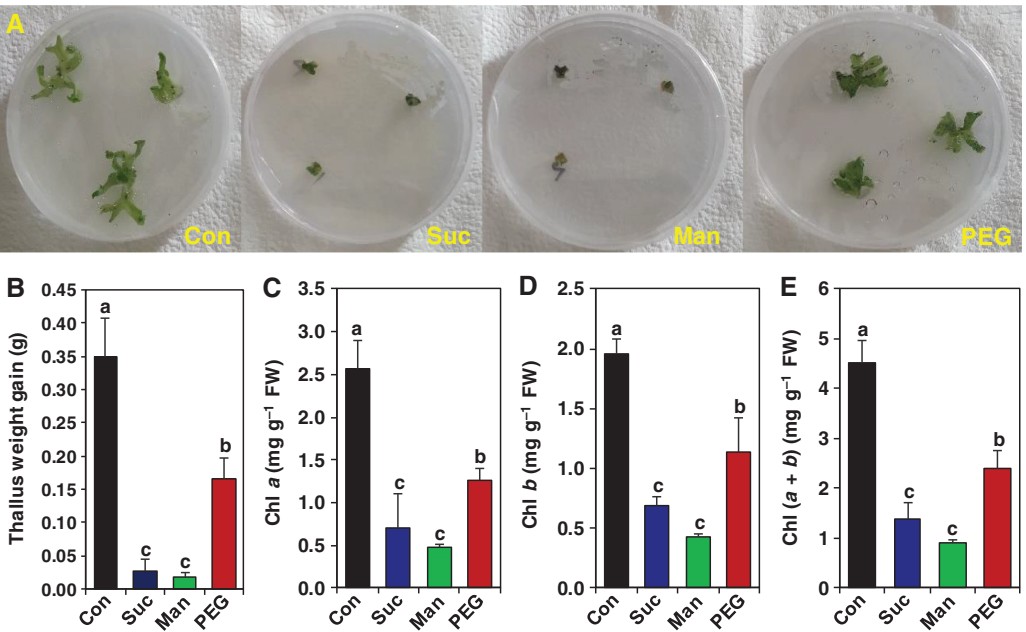

**Figure 1 Effect of drought on the growth and chlorophyll (Chl) content in *M. polymorpha*.** For growth analysis, after taking preliminary weight, the part of the 15 days aged thallus body was cultured in ½ B5 solid medium (control; Con) and ½ B5 solid medium supplemented with 0.2 M sucrose (Suc), 0.5 M mannitol (Man) and 10% polyethylene glycol (PEG). (A) The phenotype and (B) thallus weight-gain were recorded after 15 days of culture. For the analysis of chlorophyll pigments, the 3-day-cultured gemmalings were exposed to the ½ B5 liquid medium (control; Con) and ½ B5 liquid medium supplemented with 0.2 M sucrose (Suc), 0.5 M mannitol (Man) and 10% polyethylene glycol (PEG). The content of (C) Chl *a*, (D) Chl *b* and (E) Chl (*a* + *b*) was determined after 2 days of drought treatments. Error bars indicate the standard errors ($n = 3$). Different alphabetic letters above the bars show statistically significant differences among the treatments ($P < 0.001$, according to least significant difference test).

## Drought-induced increment in the levels of $H_2O_2$, lipid peroxidation product (MDA) and electrolyte leakage in *M. polymorpha*

Drought commonly induces the production of excessive ROS, including $H_2O_2$, which is involved in lipid peroxidation-mediated disintegration of membrane structure (*Sachdev et al., 2021*). To observe whether drought induces oxidative damage in *M. polymorpha*, we assessed the status of several oxidative stress indicators, including $H_2O_2$, MDA and electrolyte leakage in the gemmalings of *M. polymorpha*. In comparison with drought stress-free control, significant increase in the contents of $H_2O_2$ (by 100%, 85.71% and 63.17%), MDA (by 257.14%, 178.57% and 97.14%) and electrolyte leakages (by 176.85%, 179.88% and 60.89%, respectively) were observed in gemmalings of *M. polymorpha* exposed to 0.2 M sucrose, 0.5 M mannitol and 10% PEG, respectively (Figs. 2A–2C). These higher levels of $H_2O_2$, MDA and electrolyte leakage indicated that osmotic stresses provoked severe oxidative stress in the gemmalings of *M. polymorpha* plant. The results also suggest that liverwort *M. polymorpha* might suffer oxidative damage during their acclimation to physiological drought.

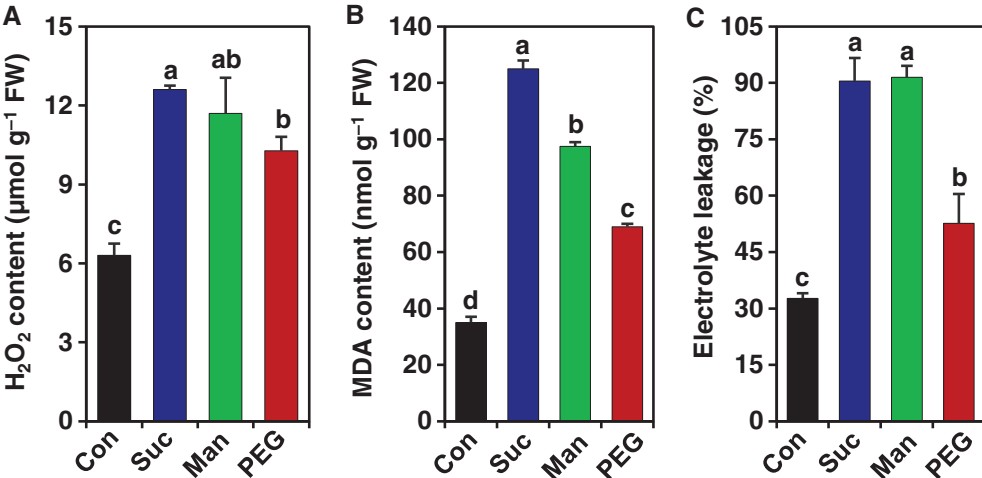

**Figure 2 Effect of drought on the levels of hydrogen peroxide ($H_2O_2$), malondialdehyde (MDA) and electrolyte leakage (%).** After 3 days of culture in ½ B5 liquid medium, the gemmalings were exposed to the ½ B5 liquid medium (control; Con) and ½ B5 liquid medium supplemented with 0.2 M sucrose (Suc), 0.5 M mannitol (Man) and 10% polyethylene glycol (PEG). The levels of (A) $H_2O_2$ and (B) MDA and (C) electrolyte leakage (%) were measured after 2 days of drought treatments. Error bars indicate the standard errors ($n = 3$). Different alphabetic letters above the bars show statistically significant differences among the treatments ($P < 0.001$, according to least significant difference test).

## Drought-induced enhancement of the activities of antioxidant enzymes in *M. polymorpha*

Plants are inherently developed with vibrant antioxidant system for scavenging excessive ROS induced by different environmental stresses, including drought (*Laxa et al., 2019*; *Mostofa et al., 2021*). Enhanced activity of enzymatic antioxidants, including SOD, CAT, APX, GST and DHAR are prerequisite for alleviating drought-induced oxidative damage (*Laxa et al., 2019*). To understand whether accumulation of ROS like $H_2O_2$ activates antioxidant defense system in *M. polymorpha*, we determined the activities of several key enzymes, such as, SOD, CAT, APX, GST and DHAR, as well as TAC in gemmalings. A significant increase in the activity of SOD (by 41.86%, 53.92% and 23.84%), CAT (by 121.33%, 137.73% and 47.71%), APX (by 130.99%, 169.01% and 47.89%), GST (by 36.98%, 30.63% and 35.56%) and DHAR (by 40.21%, 37.86% and 44.92%) was recorded in gemmalings of *M. polymorpha* subjected to 0.2 M sucrose, 0.5 M mannitol and 10% PEG, respectively, compared with those in stress-free control plants (Figs. 3A–3E). Similarly, TAC was also increased by 120.47%, 139.95% and 179.51% in the gemmalings of *M. polymorpha* stressed with 0.2 M sucrose, 0.5 M mannitol and 10% PEG, respectively, relative to control plants (Fig. 3F). This induction of enzyme activities was positively correlated with the levels of $H_2O_2$ in gemmalings of *M. polymorpha* under drought stress (Figs. 2A & 3A–3F). The results of antioxidant activities suggest that *M. polymorpha* responded to drought-induced oxidative stress by stimulating antioxidant defense system.

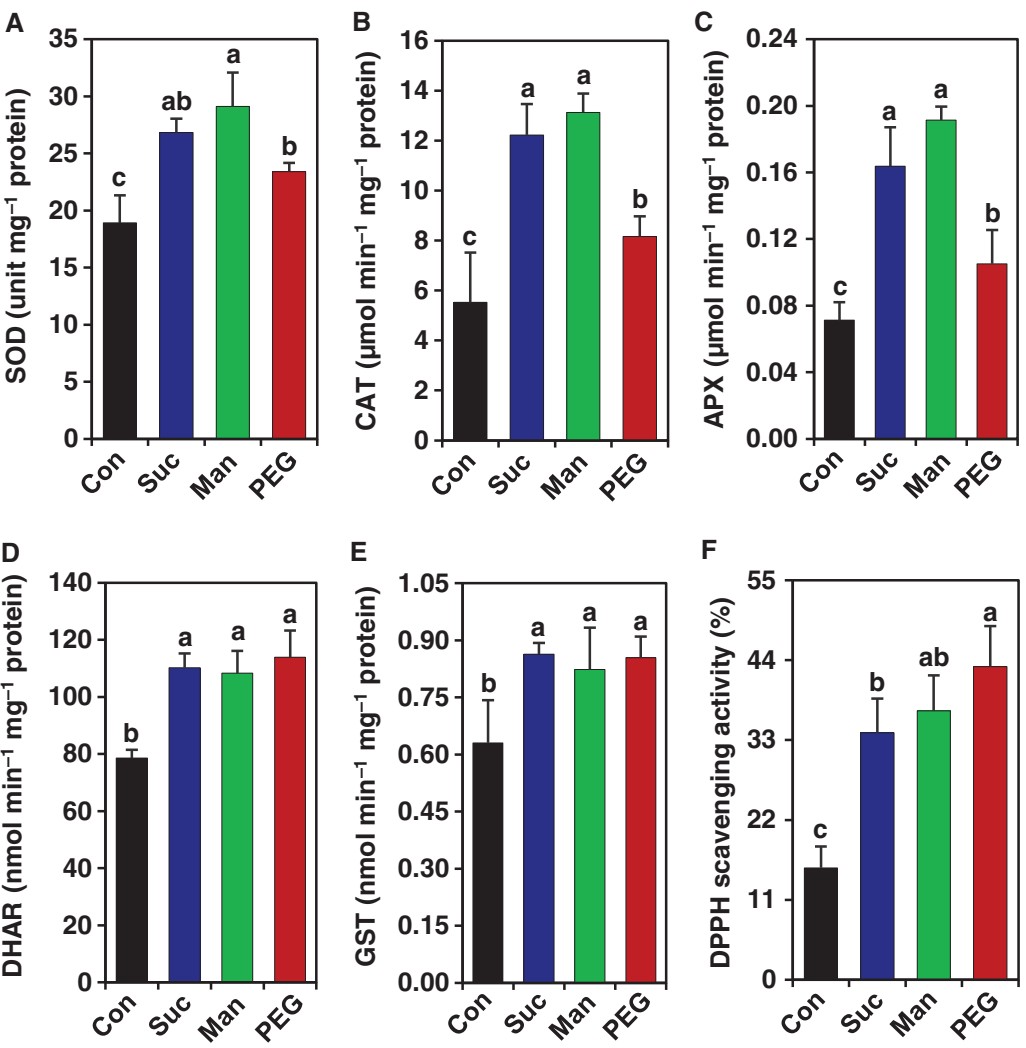

**Figure 3 Effect of drought on the antioxidant activity in *M. polymorpha*.** After 3 days of culture in ½ B5 liquid medium, the gemmalings were exposed to the ½ B5 liquid medium (control; Con) and ½ B5 liquid medium supplemented with 0.2 M sucrose (Suc), 0.5 M mannitol (Man) and 10% polyethylene glycol (PEG). The activities of (A) superoxide dismutase (SOD), (B) catalase (CAT), (C) ascorbate peroxidase (APX), (D) dehydroascorbate reductase (DHAR) and (E) glutathione *S*-transferase (GST) were determined after 2 days of drought treatments. (F) Total antioxidant capacity (DPPH scavenging activity (%)) was determined from the methanolic extracts of control and drought-treated gemmalings. Error bars indicate the standard errors ($n = 3$). Different alphabetic letters above the bars show statistically significant differences among the treatments ($P = 0.001$ for figures B, C and F, $P < 0.001$ for A and D, $P = 0.049$ for E according to least significant difference test).

## Drought-induced proline accumulation in *M. polymorpha* and phylogenetic relationship of proline biosynthesis gene

Indeed, osmoprotectant accumulation is one of the paramount strategies of plant to maintain water status under water-shortage conditions. Among the osmoprotectants, proline has been found to be accumulated in different crop species upon their exposure to abiotic stresses, including drought (*Dikilitas, Simsek & Roychoudhury, 2020*). In the

current study, we were interested to evaluate the relationship between accumulation of proline and drought stress in the liverwort *M. polymorpha*. In relation to the stress-free control, the level of proline was increased by 561.11%, 640.74% and 453.70% in the gemmalings of *M. polymorpha* exposed to 0.2 M sucrose, 0.5 M mannitol and 10% PEG, respectively (Fig. 4A). Because phylogenetic analysis enriches our understanding in the evolutionary relationship among genes, genome and species, we performed phylogenetic tree analysis of proline biosynthesis gene (*P5CS1*) to identify how proline is evolutionarily linked to land plants' adaptations to osmotic stress. We built a phylogenetic tree using protein sequences of P5CS1 of widely accepted model plants, such as, bryophytes (mosses *C. purpureus*, *P. patens*, *S. fallax*, and *S. magellanicum*, liverwort *M. polymorpha*), pteridophyte (*C. richardii*), lycophyte (*S. moellendorffii*) and angiosperms (*A. thaliana*, *O. sativa* and *P. trichocarpa*). The greater extent of sequence similarity by ClustalW program suggested that the P5CS1 is likely to be highly conserved throughout the land plants (File S1). The branching behavior of tree indicated that proline biosynthesis gene *P5CS1* was evolutionarily shared by the bryophytes, including liverwort *M. polymorpha*, lycophytes, pteridophytes and angiosperms (Fig. 4B). Taken together, the enhanced level of proline accumulation in *M. polymorpha* and the output of phylogenetic analysis suggest that osmolyte proline might be an integral part of osmotic stress adaptation strategy in *M. polymorpha*, and are likely to be involved in other land plants' adaptations to dry habitats.

## Involvement of ABA in the alleviation of drought effects on *M. polymorpha*

ABA is ubiquitous plant hormone, and has been considered as a potential regulator of a wide range of cellular processes associated with plant growth, development and abiotic stress responses and tolerance (*Sakata, Komatsu & Takezawa, 2014*). To assess the role of ABA in drought stress mitigation in *M. polymorpha*, we applied exogenous ABA to the gemmalings prior to their exposure to 0.2 M sucrose, 0.5 M mannitol and 10% PEG, and subsequently evaluated their tissue damage under osmotic stressed conditions.

The viability of the thallus tissues of *M. polymorpha* was examined using Evan's blue uptake test. The microscopic observation showed higher survivability of the gemmalings pretreated with ABA compared with ABA-untreated gemmalings (Fig. 5A). Next, we were interested in exploring whether ABA could play any role in the activation of antioxidant system and acceleration of proline accumulation in the gemmalings of *M. polymorpha*. In comparison with the control, ABA-treated gemmalings showed significantly enhanced activity of SOD (by 38.37%), CAT (by 253.18%), APX (by 200.00%) and DHAR (by 30.73%), as well as TAC (by 134.24%); however, the upregulation of GST activity remained comparable with the control plants (Figs. 5B–5G). On the other hand, proline content was remarkably improved by 338.89% in ABA-treated gemmalings, when compared with that in ABA-untreated gemmalings (Fig. 5H). These findings clearly indicated that ABA might play a pivotal role in induction of antioxidant system and osmoprotectants to mitigate the adverse effects of drought in liverwort *M. polymorpha*.

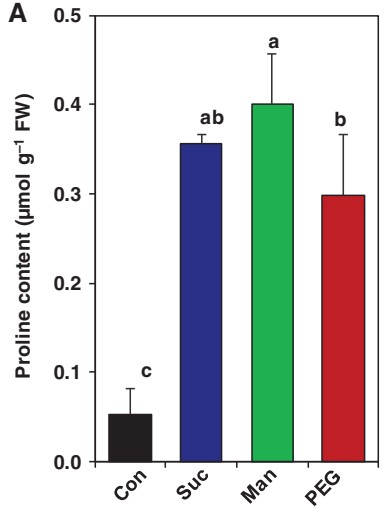

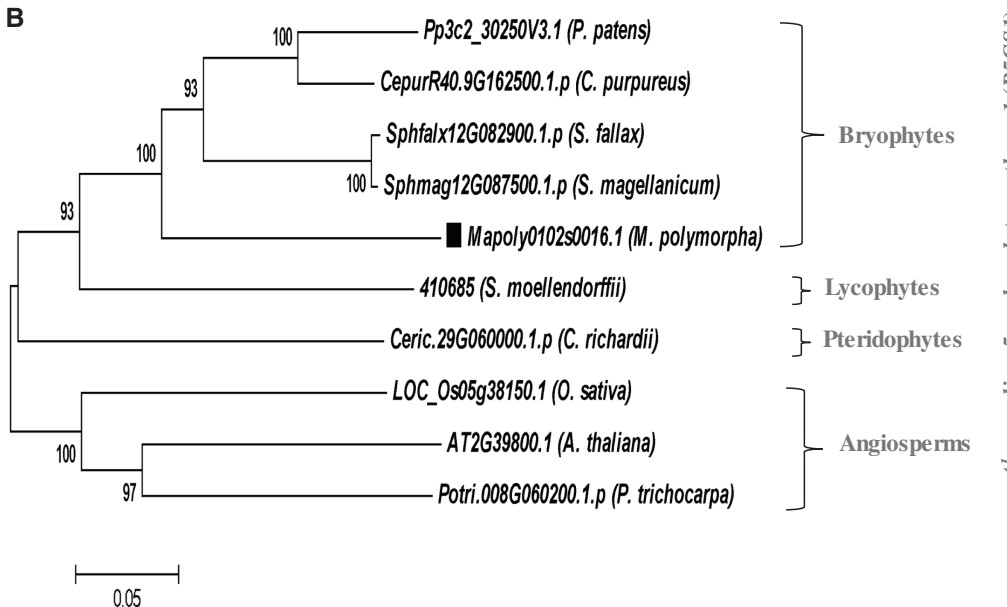

**Figure 4 Effect of drought on the proline content in *M. polymorpha* and phylogenetic analysis of proline biosynthesis gene.** After 3 days of culture in ½ B5 liquid medium, the gemmalings were exposed to the ½ B5 liquid medium (control; Con) and ½ B5 liquid medium supplemented with 0.2 M sucrose (Suc), 0.5 M mannitol (Man) and 10% polyethylene glycol (PEG). (A) The proline content was determined after 2 days of drought treatments. Error bars indicate the standard errors ($n$ = 3). Different alphabetic letters above the bars show statistically significant differences among the treatments ($P$ = 0.001, according to least significant difference test). (B) Phylogenetic analysis of proline biosynthesis gene *P5CS1* in bryophytes (Moss *C. purpureus*, *P. patens*, *S. fallax*, and *S. magellanicum*, liverwort *M. polymorpha*), pteridophyte (*C. richardii*), lycophyte (*S. moellendorffii*), and angiosperms (*A. thaliana*, *O. sativa* and *P. trichocarpa*). Protein sequences were collected by blast search in Phytozome v13 databases. The alignment of the sequences was made by ClustalW program and the tree was built using the Maximum likelihood method. The number on the branches indicates bootstrap values (1,000 replications). Bar indicates the number of substitutions per site.

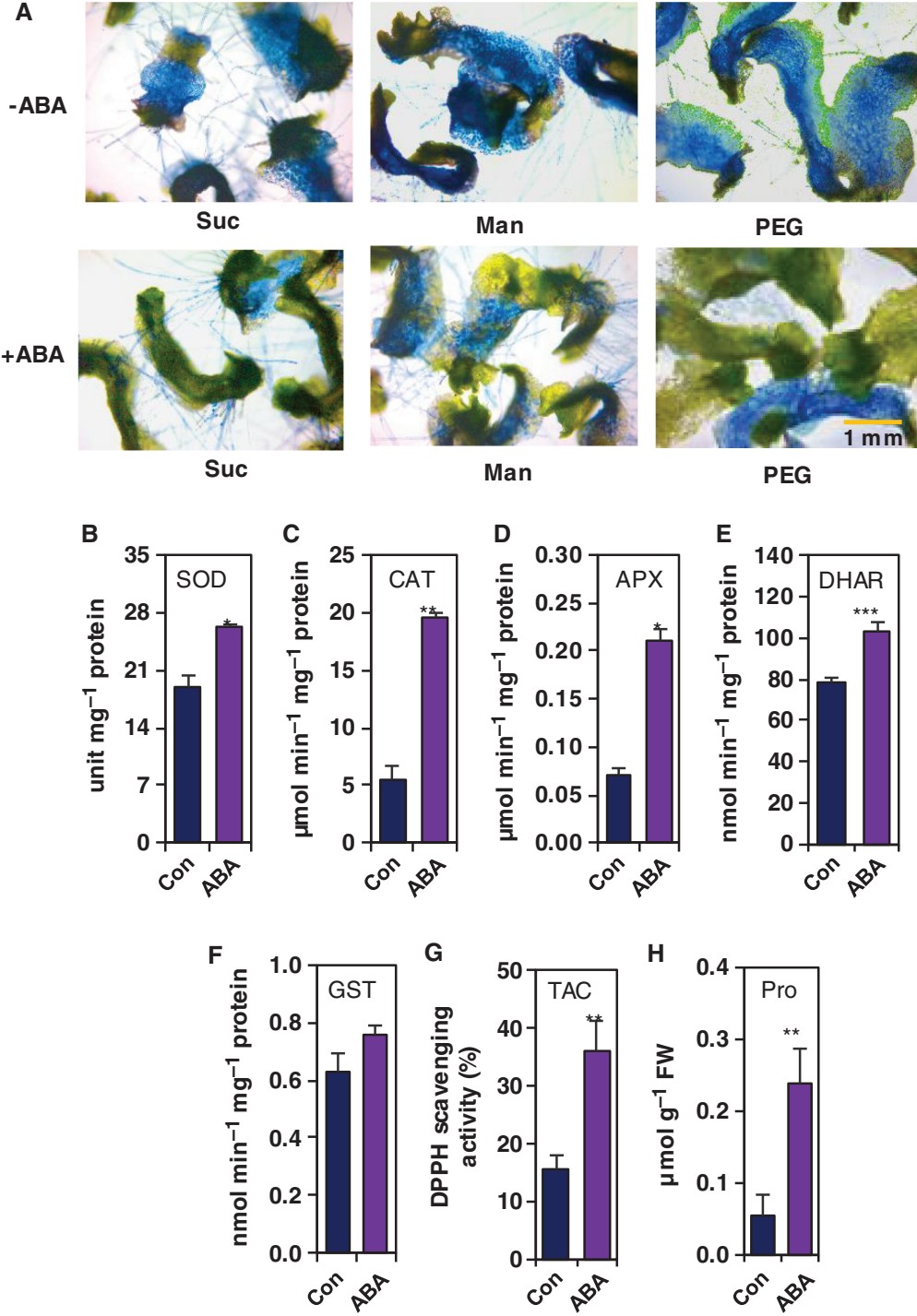

**Figure 5 Effect of ABA for alleviating drought-induced tissue damage and induction of antioxidant activity and proline content.** The gemmalings pretreated with and without ABA (10 μM) for 2 days were exposed to the ½ B5 liquid medium supplemented with 0.2 M sucrose (Suc), 0.5 M mannitol (Man) and 10% polyethylene glycol (PEG). (A) The tissue damage induced by drought conditions was observed after 2 days of treatments using Evan's blue staining. For showing antioxidant activity and proline content, the 3-day-cultured gemmalings were exposed to the ½ B5 liquid medium (control; Con) and ½ B5 liquid medium supplemented with ABA (10 μM). The activities of (B) superoxide dismutase (SOD), (C) catalase (CAT), (D) ascorbate peroxidase (APX), (E) dehydroascorbate reductase (DHAR) and (F) glutathione

**Figure 5** (continued)
*S*-transferase (GST), (G) total antioxidant capacity; TAC (DPPH scavenging activity (%)) and (H) proline (Pro) content were measured after 2 days of ABA treatments. Error bars indicate the standard errors (*n* = 3). Asterisks indicate significant differences between the treatments (*P* = 0.011, 0.002, 0.022, 0.001, 0.142, 0.004, 0.005 for figures B, C, D, E, F, G and H, respectively). *$*P \leq 0.05$, **$*P \leq 0.01$, ***$*P \leq 0.001$, according to Student's t-test.

## DISCUSSION

During acclimation to drought stress, land plants undergo numerous changes at their cellular levels, leading to the alteration of morphology and physiology to support their survival against water-scarcity. For instance, mosses concealed their surface area, reorganized their structures and formed unique features like brachycytes in responses to drought and ABA (*Bhyan et al., 2012*; *Takezawa et al., 2015*; *Arif et al., 2019*; *Ríos-Meléndez et al., 2021*). The moss *Sanionia uncinata* reduced volume of phyllids and cauloids during their adaptation to dry conditions (*Pizarro et al., 2019*). However, in contrast to angiosperms, bryophytes were less commonly explored in terms of their growth and survival responses against drought. In the present study, physiological drought induced by sucrose, mannitol and PEG treatments led to significant growth inhibition in *M. polymorpha*, as evidenced from their poor phenotypes and FW reduction (Figs.1A & 1B). Similar to our findings, *M. polymorpha* gemmalings exhibited severe growth defects with structural deformity following their exposure to limited level of water and salinity stress (*Tanaka et al., 2018*; *Godinez-Vidal et al., 2020*). These results suggest that *M. polymorpha* like other land plants also suffered from growth retardation when they were exposed to osmotic-related stresses like salinity and drought, which might be a common response of land plants irrespective of their evolutionary origins.

Reduction of the levels of photosynthetic pigments is a general consequence of drought stress in angiosperms (*Liu et al., 2018*; *Fallah, 2020*). The possible reasons for decreasing chlorophyll content are generally linked to the inhibition of photosynthetic pigments synthesis and deterioration of photosynthesis apparatus like thylakoid membranes due to the perturbation of redox balance under water-limiting conditions. Our data also revealed that drought stress caused a significant reduction in the contents of Chl *a*, Chl *b* and Chl (*a* + *b*) (Figs. 1C–1E), which positively coincided with the biomass reduction in *M*. polymorpha (Figs. 1A & 1B). In accordance with our study, the drought and osmotic stress-induced negative consequences on photosynthetic efficiency were also recorded in mosses (*Proctor & Pence, 2002*; *Pizarro et al., 2019*; *Ćosić et al., 2020a*). These results indicated that maintenance of photosynthetic pigments was indispensable for maintaining better growth during the adaptation of land plants to the dry terrestrial habitats.

Drought provokes ROS accumulation in plants by disrupting electron transport system. Elevated levels of ROS in land plants trigger membrane lipid peroxidation, causing destruction of membrane integrity and the ultimate loss of electrolytes through damaged membranes (*Guo et al., 2018*). The present study found that drought imposed by sucrose, mannitol and PEG resulted in a significant accumulation of $H_2O_2$ and MDA in the gemmalings of *M. polymorpha* (Figs. 2A & 2B). The greater electrolyte leakage in

*M. polymorpha* also positively correlated with the contents of lipid peroxidation product MDA under drought conditions (Figs. 2B & 2C). These results indicated that drought induced an oxidative stress in *M. polymorpha*, which possibly contributed to the negative effects of drought. In parallel to our findings, enhanced accumulation of MDA and a higher degree of electrolyte leakage were reported under dehydration stress in thalloid liverwort *Monoclea forsteri* (*Hooijmaijers, 2008*). A positive correlation between MDA content and electrolyte leakage were also found in another basal desiccation-tolerant moss, *Atrichum undulatum*, under dehydration conditions (*Hu et al., 2016*).

To confront drought stress, plants employ various defense strategies, such as activation of ROS scavenging systems, elevation of osmoprotectant levels and protection of membrane structure (*Khaleghi et al., 2019*). In particular, stimulation of antioxidant potential in land plants is an integral defense strategy to combat ROS-toxicity under drought stress (*DaCosta & Huang, 2007*; *Laxa et al., 2019*). As a first line of enzymatic defense against ROS, SOD actively converts $O_2^{\bullet-}$ into $H_2O_2$. The relatively less active $H_2O_2$ is then converted into $H_2O$ with the help of other antioxidant enzymes, including CAT and APX. The coordination among the antioxidant components is crucial to ensure better protection of cells from excessive ROS-caused oxidative damage (*Ren et al., 2021*). Overexpression of *SaCu/Zn SOD* from *Sedum alfredii* in *A. thaliana* enhanced the resistance against oxidative stress (*Li et al., 2017*). Alongside, the increased activity of SOD in *Hordeum vulgare* under heat and drought stress (*Zhanassova et al., 2021*) indicated that the induction of SOD activity might be crucial to alleviate oxidative stress induced by drought and/or heat stress in angiosperms. Similar to those, SOD activity was also found to be significantly increased in *Selaginella tamariscina*, *Calypogeia granulate*, *Barbula unguiculata* and *Plagiomnium cuspidatum* and *S. uncinata* in response to different abiotic stresses (*Nakagawara et al., 1993*; *Nakata et al., 2002*; *Wu et al., 2009*; *Wang et al., 2010*; *Salekjalali, Haddad & Jafari, 2012*; *Pizarro et al., 2019*). Our results on SOD activity (Fig. 3A) are in supports of the previous studies, which indicated that the induction of SOD activity might be a strong defense response in liverworts against drought stress. We also recorded a heightened activity of CAT in drought-exposed *M. polymorpha* (Fig. 3B), which supports the results observed in lycophyte *Selaginella tamariscina*, moss *S. uncinata and* angiosperm *H. vulgare* under drought stress (*Wang et al., 2010*; *Pizarro et al., 2019*; *Zhanassova et al., 2021*). In addition to CAT, the ascorbate-glutathione (AsA-GSH) cycle enzyme APX plays pivotal roles in detoxification of $H_2O_2$ in angiosperms (*Smirnoff & Arnaud, 2019*). Overexpression of *CaAPX* from *Camillia azalea* in tobacco enhanced cold and heat tolerance, whereas *AgAPX* from *Apium graveolens* conferred drought tolerance in transgenic *A. thaliana* (*Wang et al., 2017*; *Liu et al., 2019b*). These results indicated the potential roles of APX in abiotic stress tolerance in angiosperms. Similar to angiosperms, APX was found to be involved in the removal of $H_2O_2$ in moss *Brachythecium velutinum* and liverwort *M. polymorpha* (*Paciolla & Tommasi, 2003*). The APX activity was also found to be enhanced in liverwort *Calypogeia granulata* by abiotic elicitor vanadate (*Nakagawara et al., 1993*), in aquatic bryophytes *Fontinalis antipyretica* by heavy metals (*Dazy et al., 2008*) and in *S. uncinata* by desiccation stress (*Pizarro et al., 2019*). The enhanced level of APX activity observed in the current
study (Fig. 3C) also supported those findings. We also analysed the activity of DHAR, which is another key component of AsA-GSH cycle and have been reported to be the part of acclimation processes to various abiotic stresses, including drought in angiosperms (*Khaleghi et al., 2019*). The increased DHAR activity (Fig. 3D) pointed out that *M. polymorpha* might have employed DHAR to regenerate AsA and maintain AsA-related redox status under drought-induced oxidative stress. GST is another ubiquitous member of the antioxidant defense system, playing substantial roles in plant tolerance to various abiotic stresses (*Dixon, Skipsey & Edwards, 2010*). Increased GST activity was found to be effective in counteracting cadmium toxicity in moss *Leptodictyum riparium* (*Bellini et al., 2020*). In the current study, the increased activity of GST in *M. polymorpha* (Fig. 3E) suggested that GST might have played crucial role during the earlier adaptation of land plants. Collectively, the increased activities of SOD, CAT, APX, DHAR and GST in *M. polymorpha* under physiological drought, and the enhanced transcript level of differentially expressed genes encoding SOD, CAT, APX, and DHAR in *Arabidopsis* under salt, cold, light and heat stresses (*Filiz et al., 2019*) suggest that the increased activities of these enzymes are consistently conserved throughout the land plants, including liverworts, for conferring resistance to oxidative stress under a wide range of abiotic stresses. Furthermore, the heightened DPPH free radical scavenging ability (Fig. 3F) coincided with the increased activities of enzymatic antioxidants (Figs. 3A–3E), implying that the liverwort *M. polymorpha* successfully stimulated the antioxidant system for alleviating excessive ROS-mediated adverse effects under drought.

Another important strategy of drought acclimation involves the maintenance of osmotic adjustment through the accumulation of various organic osmolytes. Among the osmolytes, proline is one of the most critical cellular protectants found to be involved in maintaining osmotic adjustment, membrane stability, scavenging ROS and cellular homeostasis (*Patade, Lokhande & Suprasanna, 2014*). Proline accumulation was also found to be greatly enhanced in angiosperm rice following its exposure to drought and osmotic stresses (*Mishra et al., 2018*; *Saha, Begum & Nasrin, 2019*; *Saddique et al., 2020*). Along with angiosperms, the enhanced level of proline accumulation was reported in mosses *A. undulatum* and *Pseudocrossidium replicatum* in response to drought stress (*Liang et al., 2013*; *Hu et al., 2016*; *Ríos-Meléndez et al., 2021*). In our observation, enhanced accumulation of proline in liverwort under physiological drought conditions (Fig. 4A) suggested that the cellular osmoprotectant machinery was crucial for the adaptation of liverworts to osmotic-related stresses. Indeed, the phylogenetic analysis of proline biosynthesis gene *P5CS1* in the representative models of land plants indicated that *P5CS1* was evolutionarily shared by the land plants, including liverwort *M. polymorpha* (File 1 & Fig. 4B).

ABA plays diverse roles in drought tolerance by regulating various plant defense systems, including ROS-detoxification by antioxidant enzymes and osmotic adjustment by enhancing the accumulation of osmoprotectants in many plant species (*Planchet et al., 2011a*; *Planchet et al., 2011b*; *Liu et al., 2019a*; *Cao et al., 2020*). Along with angiosperms, the protective role of ABA against osmotic stress was also observed in different species of bryophytes (*Shinde, Islam & Ng, 2012*; *Takezawa et al., 2015*; *Saruhashi et al., 2015*;

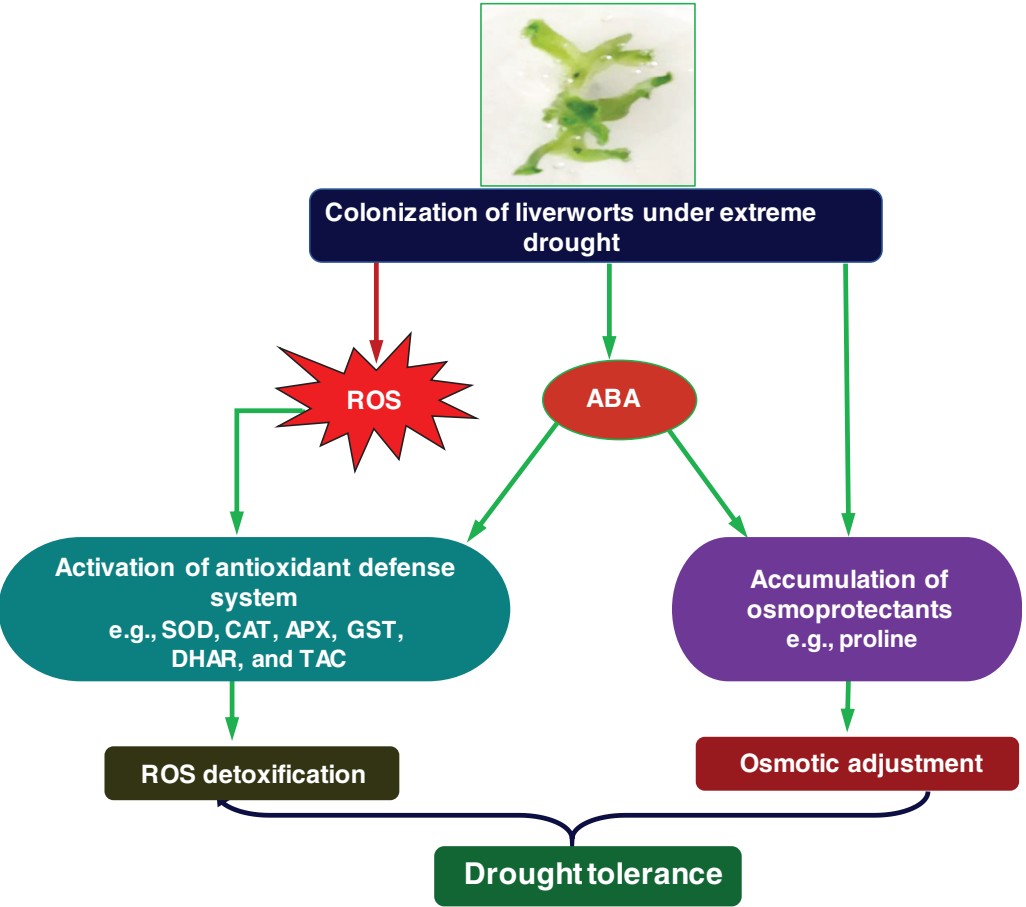

**Figure 6 Thematic model illustrating drought tolerance mechanisms of diverging land plant liverwort *M. polymorpha*.** During adaptation to the dry terrestrial habitat, liverwort *M. polymorpha* had to face extreme water-limiting conditions and suffered drought-induced stresses, including oxidative and osmotic stresses. In response, liverwort *M. polymorpha* might have engaged cellular antioxidants for detoxifying reactive oxygen species (ROS) and accumulated osmoprotectants for osmotic adjustment. Along with these, they might have enhanced abscisic acid (ABA) biosynthesis, which could also contribute to the greater accumulation of osmoprotectants and increased activity of cellular antioxidants to maintain water potential and ROS-detoxification, respectively. Thus, the mechanisms of drought tolerance led by the cellular antioxidants (SOD, CAT, APX, GST and DHAR), osmoprotectant (proline) and phytohormone (ABA) were developed in the evolutionarily important monophyletic group of land plants like liverworts. SOD, superoxide dismutase; CAT, catalase; APX, ascorbate peroxidase; DHAR, dehydroascorbate reductase; GST, glutathione *S*-transferase; TAC, total antioxidant capacity.

*Liu et al., 2019a*; *Godinez-Vidal et al., 2020*; *Wu et al., 2021*). Our data showed that ABA pretreated drought-exposed gemmalings displayed less tissue damage when compared with ABA untreated gemmalings (Fig. 5A). This result suggests that bryophytes, including liverwort *M. polymorpha* engaged ABA to increase their survivability against drought stress. Previous reports also demonstrated that ABA essentially triggered the antioxidant enzyme activities in angiosperms wheat and maize in response to drought stress (*Wei et al., 2015*; *Yao et al., 2019*). A lower level of ABA contributed to the activation of enzymatic antioxidant system in liverwort *M. polymorpha* and mosses *P. patens, A. undulatum,*

*Entosthodon hungaricus*, and *Hennediella heimii* (*Vujicic et al., 2017*; *Ćosić et al., 2020b*). The ABA-mediated enhancement of antioxidant enzyme activities was also reported in aquatic seaweed (*Guajardo, Correa & Contreras-Porcia, 2016*), indicating that the mechanisms was initiated earlier than the terrestrial plant. In the current study, the ABA-induced increase of SOD, CAT, APX, DHAR and GST activities, total antioxidant capacity, and osmoprotectant proline level in the gemmalings of *M. polymorpha* (Figs. 5B–5H) suggest that ABA was the key player in the modulation of antioxidant activity and maintaining osmotic adjustment during adaptation and evolution of land plants. Altogether, our results underpinned that during adaptation to dry habitat, liverworts might have engaged antioxidant system, osmoprotectant proline and phytohormone ABA, which collectively conferred drought tolerance in this group of land plants (Fig. 6).

## CONCLUSIONS

Based on the morpho-physiological and biochemical data, we conclude that similar to angiosperms, the gemmae of *M. polymorpha* showed growth inhibition and reduction of photosynthetic pigment contents under drought conditions. The enhanced level of $H_2O_2$ and lipid peroxidation product MDA with a higher degree of electrolyte leakage in *M. polymorpha* suggests that liverworts might have encountered harmful effects of drought in terms of ROS-induced oxidative damage during their adaptation to harsh habitat. Along with better DPPH radical scavenging activity, the significant induction of enzymatic antioxidants, such as SOD, CAT, APX, DHAR and GST in *M. polymorpha* under physiological drought indicates that the cellular mechanisms for ROS-detoxification were well-executed in liverworts during their terrestrial adaptation. The enhanced level of proline accumulation in *M. polymorpha* in response to drought and the co-existence of proline biosynthesis gene in many land plant representatives suggest an evolutionarily adaptive role of this osmoprotectant in drought tolerance. Furthermore, the ABA-mediated reduction of tissue damage and induction of cellular protectants, such as antioxidants and proline in *M. polymorpha* reveal a putative function of this phytohormone in the adaptation of liverworts to water-shortage conditions. Our current efforts made a statement that the functions of enzymatic antioxidants, proline and ABA were the part of drought tolerance mechanisms in liverwort *M. polymorpha*, which might be essential for the adaptation of land plants to osmotic stresses. However, a comprehensive study on the expression of stress-related genes and transcriptomic analysis of genes encoding cellular antioxidants and osmoprotectants are needed to clarify more about the adaptive roles of those cellular protectants in land plant adaptation to drought.

## ACKNOWLEDGEMENTS

The authors are very much grateful to Daisuke Takazawa, Professor, Saitama University for providing the experimental materials.

### Funding

This work was supported by the Bangabandhu Sheikh Mujibur Rahman Agricultural University, Bangladesh and Ministry of Science and Technology, Bangladesh. The funders had no role in study design, data collection and analysis, decision to publish, or preparation of the manuscript.

### Grant Disclosures

The following grant information was disclosed by the authors:
Bangabandhu Sheikh Mujibur Rahman Agricultural University, Bangladesh and Ministry of Science and Technology, Bangladesh.

### Competing Interests

Mohammad Golam Mostofa is an Academic Editor for PeerJ.

### Author Contributions

- Totan Kumar Ghosh conceived and designed the experiments, performed the experiments, analyzed the data, prepared figures and/or tables, authored or reviewed drafts of the paper, and approved the final draft.
- Naznin Haque Tompa performed the experiments, prepared figures and/or tables, and approved the final draft.
- Md. Mezanur Rahman performed the experiments, analyzed the data, prepared figures and/or tables, and approved the final draft.
- Mohammed Mohi-Ud-Din analyzed the data, authored or reviewed drafts of the paper, and approved the final draft.
- S. M. Zubair Al-Meraj performed the experiments, prepared figures and/or tables, and approved the final draft.
- Md. Sanaullah Biswas conceived and designed the experiments, authored or reviewed drafts of the paper, and approved the final draft.
- Mohammad Golam Mostofa conceived and designed the experiments, analyzed the data, prepared figures and/or tables, authored or reviewed drafts of the paper, and approved the final draft.

### Data Availability

   The raw data is available in the Supplemental File.

### Supplemental Information

Supplemental information for this article can be found online at http://dx.doi.org/10.7717/peerj.12419#supplemental-information.

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
