# Peer review of "Acclimation of liverwort Marchantia polymorpha to physiological drought reveals important roles of antioxidant enzymes, proline and abscisic acid in land plant adaptation to osmotic stress"

_PeerJ, doi:10.7717/peerj.12419_

## Round 0.1 · original submission · Major Revisions

As you will see from the reviews, both reviewers appreciate your description of drought stress in Marchantia and I agree with them that this is a worthy endeavor and could be of broad interest to the field of research on land plant evolution if novel concepts were presented. However, there is substantial criticism by both reviewers on the motivation and presentation of the data and also on the experimental support and statistical analysis for some of your conclusions. Therefore I would like to ask you to carefully rewrite and revise your manuscript before resubmission.

·

Basic reporting

no comment

Experimental design

In this manuscript, Ghosh et al. studied the response of the liverwort Marchantia polymorpha to physiological drought.

I have a major concern regarding the research question and the way it is presented. The author use here Marchantia polymorpha as a model and present it as "the first land plant" which is absolutely incorrect. From this hypothesis the author conclude that their finding in Marchantia polymorpha represents the ancestral state in embryophytes, which is again incorrect and misleading.

I suggest the authors to entirely rewrite the manuscript for that important aspect. A good read before rewriting would be Delaux et al. 2019 (10.1016/j.cub.2019.09.044).

For instance:
- "diverging land plants" I assume the authors mean "early diverging land plants", which is actually a term to avoid as early diverging as to be used as a comparison with a "later diverging" group. This is irrelevant here
- "Basal land plants" cannot be used either.
- l 77-78: "bryophytes.. are considered the first land plants" this is wrong. The first land plants died more than 400 million years ago. bryophytes are as "evolved" as flowering plants or ferns. Please reformulate.
- l382: "..in basal representatives and today's land plants". This sentence would suggest that marchantia polymorpha is actually a fossil.

Material and method
"we searched the gene in all representative of terrestrial plants" please rephrase as the BLAST search was restricted to less than 10 species.

Validity of the findings

Given that I am not an expert in the physiological measurement conducted by the authors it is difficul for me to comment on that.

The phylogenetic analysis conducted on the proline biosynthesis gene seems reliable, although more details on the substitution mode etc.. would be an important addition.

As stated above, concluding on the ancestral character based on Marchantia polymorpha is not relevant. The authosr should compare with angiosperms/vascular plants to draw evolutionary hypotheses.

I would find interesting to compare the results from this paper with the recent paper by Wu et al. 2021 (10.1038/s41477-021-00929-7) that studied another stress (salt) in Marchantia polymorpha.

Reviewer 2 ·

Basic reporting

Ghosh et al treats gemmae and gemmalings of the liverwort Marchantia polymoprha with one concentration each of sucrose, mannitol, PEG and ABA. After some time, the phenotypic and molecular changes are interpreted as drought responses. Growth, chlorophyll levels, enzyme activities, etc are measured.

First of all, this manuscript needs language editing. Many times in the manuscript editing is needed for clarity, as I am not sure what the authors try to say.

Second, the framing of the study in the introduction is very poor. The introduction lacks new references. There must be lots of drought studies in liverworts and mosses that would help the authors frame their own work. Please cite them appropriately.

The results section is very short considering there are 5 figures with original data. This section could be expanded by clearly stating what was done and how, plus including references, and maybe also better motivating why some experiments were done.

The conclusions and speculations outreach the limited results presented. The authors need to read published papers of relevance for their study and cite them as appropriate throughout the manuscript.

The discussion is much longer than the results, and repeats much of the results. Present data in the results and put them in context in the discussion.

Experimental design

The authors use Suc, Man and PEG because other papers have used these treatments. If all three works as “drought” why is there a significant difference between PEG and the other two treatments in many of the experiments?

I don’t like that the authors grow plants on sugars and then write about drought treatments. There should be plenty of water in the B5 plates, and some of the experiments is done in liquid media. Wouldn’t it be better to write about sucrose, mannitol and PEG treatments? The authors can explain and discuss why their experiments should be interpreted as “drought treatments” in the discussion.

Are the plants alive after Suc and Man treatments in fig 1A? They look dead. Would they start growing normally if transferred to a normal B5 plate with nothing added?

What accessions and sexes have been used in this study? Have a single accession and sex ( = a single genotype) been used for all experiments? If so, ABI1 is reported to be located on the sex chromosomes and therefor there could be differences in ABA-mediated stress-responses between sexes in M. polymopha.

The tree is made using NJ. Not sure that method is good enough, especially considering that the tree looks poorly made (it has very low support values on many nodes). Is the tree rooted? Much more details on tree construction is needed in the M&M.

The authors use only 3 replicates in most of their experiments. I am not sure growing three genetically identical plants (clones) next to each other on the same plate actually counts as three biological replicates. Maybe there is something with that specific plate that affects the experiment. I would probable treat this set up as three technical replicates. Having clones on different plates, or even better having different genotypes of wt on the same plate, using three identical plates, would be my biological replicates if I would do this type of experiment.

I am concerned about the photo of “normal” plants in Figure 1A and Figure 6. These plants do not appear to grow normally. This looks a lot like a plant that is stressed or is receiving too much blue light and too little red light (the thalli appear to be elongated and twisted?).

The authors say the plant grow for 15 days, but the plant in fig 1A must be much older than 15 days? It has bifurcated three times. I have never seen this type of growth speed for plants that grow in white light in 16/8 photoperiod at 22C.

Marchantia gemmalings growing in liquid medium/water must be water stressed. Then the authors add Suc/Man/PEG, and suddenly the plants are “drought treated”. Wouldn’t it be better to compare drought stress to no stress, rather than another type of stress.

Validity of the findings

The raw data is provided. But I don’t see much information of the statistical analyses in the text. Also see the comment below on using numbers like 123.56%.

The statistical analyses appear to be ok, even though I don’t understand why and when different methods are being used. However, there is a problem with support in the phylogenetic tree. Branches that are not supported should be collapsed. The method for calculating branch support needs to be included.

As I wrote above, I am not sure if this manuscript is using three biological replicates (says something about biological variation) or three technical replicates (says something about how good the method is). The error bars are very small, so I would assume this set up is more to be considered as technical replicates than biological replicates.

If I understand the M&M, two different statistical test have been done. When and why were the two different tests used? This should be indicated in the figure legends or results section. Also, would be good to provide the P-values, not just stating that it is significant or not significant.

Additional comments

Below are some comments on specific parts of the text:


Line 21: Why?

Line 22: What is an “evolutionary component”?

Line 26: improper use of semi colon

Line 40-42: Inaccurate interpretation of data. M. polymorpha did not exist during the early stages of plant adaptation and evolution. The authors must revise their hypothesis and use an accurate phylogeny over land plant evolution.

Line 44-45: So they were never stressed before global warming and environmental deterioration?

Line 46: “Plants” include lots of different algae, which are not sessile.

Line 53 and later: Use “land plants” instead of “plants” if you mean land plants.

Line 77-78: This sentence need more work. And extant bryophytes are not the first land plants. What is a basal land plant, and why are extant bryophytes basal? I think the literature is becoming more and more clear about the use of inappropriate terms like “basal plants”, “lower plants”, etc. It is important to use proper terminology to not confuse the reader.

Lines 78-80: Sentence need more work.

Line 81: What is an early land plant? Maybe good to not use several different terms (of which some are not appropriate to use) for the same thing.

Line 81-83. Use commas to make sentences better structured.

Line 84: Here is another “early plant”. Why not just wright “algae”, if the authors mean algae colonized land.

Line 86-87: Is this really correct. Who says this?

Line 88: Too many commas.

Line 91-93: This is not the current view on land plant phylogeny. The authors need to stop using old and outdated papers as reference. There are several very recent papers on land plan phylogeny that should be easy to find.

Line 95: Gemmae is plural. Should be “gemma cup”. The gemma cup is not the asexual reproductive unit. The gemmae inside the cup are. Not all liverworts have gemma cups or gemmae.

Line 99: There are newer references. Please use proper references.

Line 102: No, current data clearly shows liverworts are not sister group to other land plants, or the earliest representative of land plants.

Line 106: no s in the end of liverwort.

Line 146: Place the phylogenetic analysis under its own heading.

Figure 1: The control plant in figure 1A doesn’t appear to grow normally. Are the plants stressed? Something might be off with the growth conditions. Also, the plant in fig 6 looks very strange. Might be too warm or too much/little light of a specific wave length?

Figure 1: Have the dormant gemmae been taken from cups and placed directly on the plates with treatments? If so, can the authors distinguish between effects of delayed germination of gemmae and effects on growth retardation in gemmalings, in figure 1?

Figure 1: There is no time aspect, like a measurement of growth speed. Therefor it is impossible to say if this is delay of growth initiation (release of dormancy and germination) or growth retardation (slow growth from the apex of the gemmaling).

Figure 1: Why are these specific concentrations used? There is no investigation of dose effect (that the severity of the phenotype correlates to the concentration of the chemical added), and there is no reference to a previously done study of dose effect of Suc, Man and PEG on gemmae and gemmalings.

Figure 1: Wouldn’t 0,2 M sucrose be expected to have the same effect as 0,2 M mannitol?

Line 207: Is there a statistically significant difference between Suc and Man in fig. 1? If not the authors can’t say that it is.

Line 217-219: Put the results in better context. Refer back to what is known from Arabidopsis or other relevant plants

Line 233: What enzymes are these? Please write the full names at first mention. Also mention why they are relevant.

To say that something is increased by e.g. 137.76 % is very exact. Have the authors actually done their experiments in such a way that they can give numbers this exact?

Line 242: in the end of the results paragraphs the authors summarize using the wrong tempus. Our results suggest…

Paragraph starting at line 247: What is the purpose of the phylogenetic tree? Because a phylogenetic tree is not used for what the authors says it is in this paragraph.

Figure 4: The authors need to look at their tree and interpret what they see. The authors should include their alignment (which should include sequences that have been trimmed properly) as a supplemental file as this is important for the interpretation. Also, what do the different numbers in the tree show? Is a number of 30 relevant (if so, why?) or should that branch be collapsed? Why are different versions (I assume splice variants) of the same gene included in the tree? The authors need to learn how to build and present a phylogenetic tree.

Line 257: What? Also, do the authors use “e.g.” when they actually mean “i.e.”?

Line 257-259: What do the data actually say, and what is speculation? The authors must take care not to overinterpret their data, and the authors must be very clear what is speculation and what is interpretation of their data.

Line 263: “Drought exposure”… No. Exposure to different concentrations of Suc, Man and PEG.

Figure 5: The authors need to provide a reference saying that Evans Blue work on M. polymorpha tissues in the way the authors claim.

Figure 5: The photos needs to be comparable and some sort of quantitative analysis on evans blue staining should be done. The photos should be of similar brightness, zoom, etc.

Line 401, etc: I hope the authors are aware that there are previous studies showing that ABA and ABA-signaling components have a role in M. polymorpha? Cite the available literature.

---

## Round 0.2 · Major Revisions

Please follow carefully the advice of the reviewer and focus on the evolutionary context as well as a better comparisson with the existing literature.

·

Basic reporting

In this revised version the authors have clarified a number of points raised during the first round of review.

However, the main issue has not been fixed appropriately. The wording is still approximate when describing liverworts and the evolutionary context.

For instance “Bryophytes, which include mosses, liverworts, and hornworts, are a monophyletic group of the earliest land plants, with mosses and liverworts belonging to their own clade (Setaphyta) and hornworts being sister to other land plants” (L85) contradict itself. If Hornworts are sister to other land plants, then Bryophytes do not form a monophyletic clade. In the same sentence “earliest land plants” again suggests that Bryophytes are some sort of fossils.

I encourage the authors to carefully edit the manuscript. I also agree with reviewer 2 that the presented data should be better compared to the existing (recent) literature on the topic. Citing these papers is not sufficient. I appreciated for instance to see Wu et al. 2021 cited as I proposed in my previous review, however the authors should discuss their result in light of that other paper.

Experimental design

The experimental setup is now properly presented.

Validity of the findings

The presented data seems solid.

---

## Round 0.3 · accepted · Accept

Thank you very much for this second revision that addresses at least the most urgent requests from the reviewers.